# Discovery of driver non-coding splice-site-creating mutations in cancer

Song Cao[1,2], Daniel Cui Zhou[1,2], Clara Oh[1,2], Reyka G. Jayasinghe[1,2], Yanyan Zhao[1], Christopher J. Yoon[1,2], Matthew A. Wyczalkowski[1,2], Matthew H. Bailey[1,2], Terrence Tsou[1,2], Qingsong Gao[1,2], Andrew Malone[1], Sheila Reynolds[3], Ilya Shmulevich[3], Michael C. Wendl[2,4,5], Feng Chen[1,6 ✉] & Li Ding[1,2,4,6 ✉]

Non-coding mutations can create splice sites, however the true extent of how such somatic non-coding mutations affect RNA splicing are largely unexplored. Here we use the MiSplice pipeline to analyze 783 cancer cases with WGS data and 9494 cases with WES data, discovering 562 non-coding mutations that lead to splicing alterations. Notably, most of these mutations create new exons. Introns associated with new exon creation are significantly larger than the genome-wide average intron size. We find that some mutation-induced splicing alterations are located in genes important in tumorigenesis (*ATRX*, *BCOR*, *CDKN2B*, *MAP3K1*, *MAP3K4*, *MDM2*, *SMAD4*, *STK11*, *TP53* etc.), often leading to truncated proteins and affecting gene expression. The pattern emerging from these exon-creating mutations suggests that splice sites created by non-coding mutations interact with pre-existing potential splice sites that originally lacked a suitable splicing pair to induce new exon formation. Our study suggests the importance of investigating biological and clinical consequences of noncoding splice-inducing mutations that were previously neglected by conventional annotation pipelines. MiSplice will be useful for automatically annotating the splicing impact of coding and non-coding mutations in future large-scale analyses.

---

[1] Department of Medicine, Washington University in St. Louis, St. Louis, MO 63110, USA. [2] McDonnell Genome Institute, Washington University in St. Louis, St. Louis, MO 63108, USA. [3] Institute for Systems Biology, Seattle, WA 98109, USA. [4] Department of Genetics, Washington University in St. Louis, St. Louis, MO 63110, USA. [5] Department of Mathematics, Washington University in St. Louis, St. Louis, MO 63130, USA. [6] Siteman Cancer Center, Washington University in St. Louis, St. Louis, MO 63110, USA. ✉email: fchen@wustl.edu; lding@wustl.edu

arge-scale studies, such as The Cancer Genome Atlas (TCGA), have identified numerous driver mutations in coding regions using whole-exome sequencing (WES) data, but most non-coding sequences still lack characterization. Non-coding drivers have started to emerge from whole-genome sequencing (WGS) data. For instance, recurrent mutations in the *TERT* promoter have been found in gliomas, melanoma, and bladder cancers[1–3]. Importantly, these are associated with poor clinical outcomes in bladder cancer[3]. Other drivers in promoter regions and untranslated regions (UTRs) have been found in recent pan-cancer WGS studies[4–6] and by individual WGS studies on melanoma[7], breast cancer[8], and renal cell cancer[9], including mutations in *PLESHS1*, *WDR74*, and *SDHD*. A few studies have also reported how mutations functionally affect RNA splicing and impact human disease[10–13]. For example, an intronic germline mutation in *COL6A1* creates a new exon associated with collagen dystrophy[12]. However, the true extent of how such somatic non-coding mutations affect RNA splicing in tumors remains largely unknown.

The identification and analysis of non-coding events and the identification of contributing drivers are crucial, open problems in cancer genomics. Some tools, such as OncoDriveFML[14] and LARVA[15], predict mutation impact by leveraging data from ENCODE, but their general frameworks are unable to identify more specific phenomena, such as splicing perturbations. Here, we apply MiSplice (Mutation-Induced Splicing), a bioinformatics tool that can identify mutation-induced splice forms in cancer using a combination of WGS and RNA-sequencing (RNA-Seq) data[16], to investigate the splicing alteration from non-coding mutations. We systematically evaluate how somatic mutations in non-coding regions create splice alterations in 783 WGS samples and 9494 WES samples from TCGA, identifying 562 non-coding splice-site-creating mutations (nc-SCMs). Many of them reside in important cancer-related genes, such as *ATRX*, *BCOR*, *CDKN2B*, *MAP3K1*, *MAP3K4*, *MDM2*, *SMAD4*, *STK11*, and *TP53*. We also discover alternative splicing phenomena enriched in non-coding regions, such as new exon creation being the dominant splice alteration event.

## Results

### MiSplice pipeline, simulation, power assessments, and benchmark.
MiSplice is a fully automated and highly parallelized pipeline that applies analytical and statistical processes to discover mutation-induced splicing events, which has been applied to coding mutations (see "Methods")[16]. Here, we extend it to non-coding mutations. It consists of a processing module for junction discovery and three successive modules for filtering based on proximity to known polymorphic genes/junctions, coverage and supporting read requirements, and case/control comparisons (Supplementary Fig. 1). MiSplice jointly analyzes WGS and RNA-Seq data, scanning the transcriptome for statistically significant non-canonical sequence junctions supported by expression evidence. It is parameterized with a number of supporting reads ($M$), minimum quality of the reads ($Q$), mutation–junction distance ($N$), and fraction of supporting reads ($k$).

We conducted in silico simulations to estimate the sensitivity of the MiSplice pipeline under various coverage depths, mixture read fractions, and mutation distances. To further evaluate the performance of MiSplice pipeline (v1.1), we generated two simulated RNA-Seq data sets having average coverages of 100× and 200× to estimate the sensitivity of MiSplice when run using its heuristic ranking algorithm (processing module 3 in Supplementary Fig. 1). Specifically, we mixed reads generated from altered junction templates with reads from reference junction templates at various ratios, resulting in nine test sets

having read mixture fractions ranging from 0.1 to 0.5 in increments of 0.05. We then used MapSplice[17] for alignment. A benchmark test shows that MiSplice can identify non-canonical junctions with sensitivity >90%, given 200× coverage for the non-canonical junctions and read mixture fraction >0.2 (Fig. 1a). More broadly, at 200× coverage, MiSplice's sensitivity ranges from 0.74 to 0.97 as the mixing ratio changes from 0.1 to 0.5. As coverage increases, we observe an appreciable increase in sensitivity, resulting from the overall higher number of junction-spanning reads.

In processing module 3, MiSplice can alternatively evaluate candidate sites using a binomial hypothesis test. Here, the null hypothesis is that the junction allele fraction (JAF), which is the ratio of the number of reads supporting a site versus total number of reads at that genomic location, is indistinguishable from a value obtained by chance. Power estimates (see "Methods") can be used to quantify the necessary read depth for a given effect size. Figure 1b shows curves for standard 80%, 90%, and 95% power thresholds plotted as functions of alternative JAF versus a null hypothesis with JAF of 5%. MiSplice is well powered given sufficient coverage, for example, requiring ~150× coverage for detecting 10% JAF events at 80% power. Following these assessments, we used MiSplice to examine 783 samples with WGS mutation data and 9494 samples with WES, finding 562 nc-SCMs (Fig. 1c); see Data availability.

Compared to previous studies[16,18], the current study focuses on the comprehensive discovery of nc-SCMs. We removed coding mutations annotated as Splice_Site, Missense, Nonsense, Nonstop, In_Frame, and Frame_Shift to focus on non-coding mutations. We compared nc-SCMs in the present study and three previous studies[13,16,18], finding that 150 nc-SCMs from WGS and 178 nc-SCMs from WES are unique and not reported in these previous studies. Our previous work[16] focused on coding-region mutations and has little overlap with the current study (5% overlapping with nc-SCMs from WGS data and 11% overlapping with nc-SCMs from WES). Shiraishi et al.[18] discovered splice-associated variants (SAVs) from WES data using SAVnet and their results show ~44 and 9% overlap with nc-SCMs from WES and WGS data in the present study, respectively. The PCAWG Consortium[13] also used SAVnet to discover SAVs from WGS, with respective overlaps to this study of ~5 and 33%. Overall 66 and 48% of nc-SCMs from respective WGS and WES data sets are uniquely reported in this study.

We further estimated the sensitivity and specificity for MiSplice[16] and SAVnet[18] based on the nc-SCMs found in TCGA WGS samples in the current study and the previous study[13]. Manual review by Integrative Genomics Viewer (IGV) reveals 111 true hits over 240 nc-SCMs for SAVnet and 228 true hits over 281 nc-SCMs from MiSplice (see "Methods" and the following section), of which 79 nc-SCMs are common to both pipelines. Seventy-six out of the 79 nc-SCMs are true hits. Based on these numbers, we estimated that the sensitivity and specificity for SAVnet are ~46% and 42%, respectively, and for MiSplice are 87% and 81%, respectively. For nc-SCMs uniquely reported by SAVnet, we observed a high false-positive rate of ~78%. For the 79 nc-SCMs reported by both SAVnet and MiSplice, the false-positive rate was very low (~4%), suggesting more confident calls when reported by both tools. For the unique calls reported by MiSplice, we observed a high false-positive rate (25%). The observed difference in the nc-SCM calls reflects underlying differences in strategy and algorithm design between SAVnet and the MiSplice pipeline. SAVnet aims to predict different splicing-associated events, including exon skipping, intron retention, and splice-site-creating events. Conversely, MiSplice is focused on splice-site-creating events. Also, for the detection of SCMs, SAVnet restricts somatic variants to local positions of newly

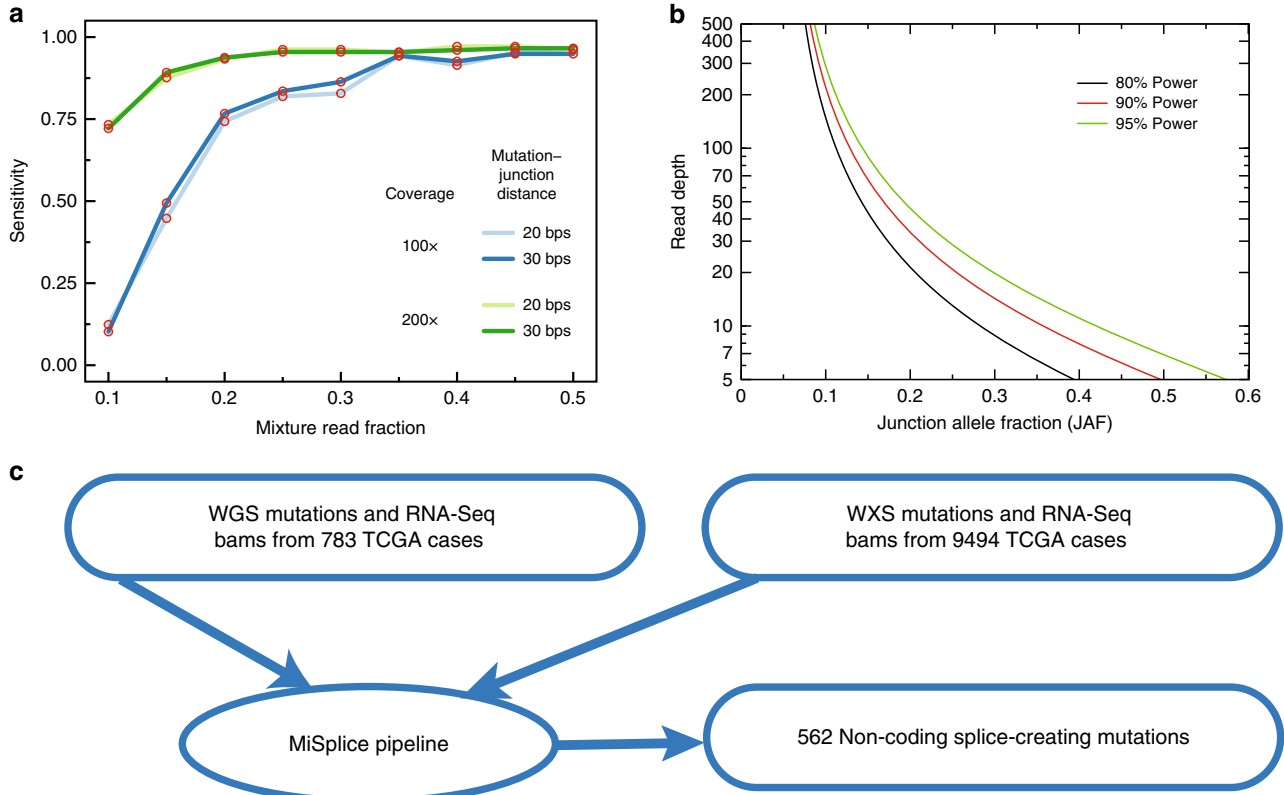

**Fig. 1 Benchmark test of MiSplice pipeline and sample set. a** Sensitivity curves derived by simulating various coverage levels and mutation distances plotted as a function of splice allele fraction (ratio of non-canonical junction reads over canonical junction reads). **b** Required junction read coverage to achieve sufficient power. **c** Sample data set processed by MiSplice pipeline. Source data are provided as a Source Data file.

created splice site or canonical splice site, that is, −3 to +6 for donor and −1 to +6 for acceptor[18]. MiSplice is designed for broader positional scope, namely, within the 20-bp window of the splice site that typically affects the splice score[16]. This larger search space enables MiSplice to detect more nc-SCMs that fall into other splicing regions near the splice site. Through this comparison, we noticed that some newly created splice sites could be far away from the mutation when that mutation is close to the canonical splice site, such that the canonical splice site is disrupted. SAVnet captures this scenario as long as the mutation is close to the canonical splice site. However, since MiSplice implements a 20-bp cut-off between the mutation and the newly created splice site, it misses these unique calls predicted by SAVnet, which fall in this category. In addition, in MiSplice, we incorporated a method for calculating splice scores, which enables improved characterization of these newly created splice sites.

**Discovery of nc-SCMs from 783 TCGA whole-genome data.** We applied MiSplice to 783 TCGA samples having both WGS and RNA-Seq data (Fig. 1c) to identify non-coding mutations leading to splice alterations. Here, we chose $M = 5$, $Q = 20$, $N = 20$, and $k = 0.05$, based on power analyses (see "Methods"). We found 281 non-coding mutations associated with splice alterations, which were manually reviewed with IGV[19]; see "Methods." Each splice junction had at least five supporting reads. In addition, sites with spliced-in mutations were required to have at least 30% of reads supporting the junction. Consistent with our previous publication, we defined spliced-in mutations as mutations that were found in RNA-Seq reads supporting the junction[16]. The heuristic 30% cut-off that we set is to further support the association between mutation and the junction in addition to the case and control test. For spliced-in mutations, we also added a

specific manual review to check that reads, which cover both junction and mutation site, contain the specific mutation; see "Methods." Two hundred and twenty-eight sites passed these manual review criteria (Supplementary Data 1), of which 189 were found to create splice forms in intronic and UTR regions, 14 in 5′ flanking regions, 6 in RNAs, and 7 in intergenic regions (IGRs) (Fig. 2a, b). These manual review statistics suggest a specificity of 81%.

We further categorized these events into seven groups according to their splicing impact: new exon creation, exon extension, exon shrinkage, exon splitting, gene fusion, new transcript, and complex events (Fig. 2a, b). Exon creation is especially notable since it has been rarely reported previously[13], but here accounts for 109 of the total 228 events (47%). Exon extension follows with 71 events. We also detected 22 new transcript, 9 exon splitting, 6 complex, 8 exon shrinkage, and 3 fusion events, where the complex events category is defined as a combination of multiple events, such as exon shrinkage and new exon (Fig. 2a, b). We found that the lengths of introns in which exon extension and shrinkage occur are comparable to the genome-wide average (~3000 bp) (Fig. 2c). However, introns with exon-creating mutations are generally an order of magnitude larger, suggesting that longer introns are appreciably more biased toward new exon creation (Fig. 2c and "Methods").

**Characteristics of non-coding splice alterations.** We used MaxEntScan[20] to estimate the splice strength of the reference and mutant splice sites ("Methods"). For both donor and acceptor sites, we observed equal or higher splicing scores in the junctions with mutations compared to the reference (mean scores of 3.48 and 0.88, respectively) (Fig. 3a). On two occasions, we found sites with relatively low splicing scores, but these are attributed to the usage of non-

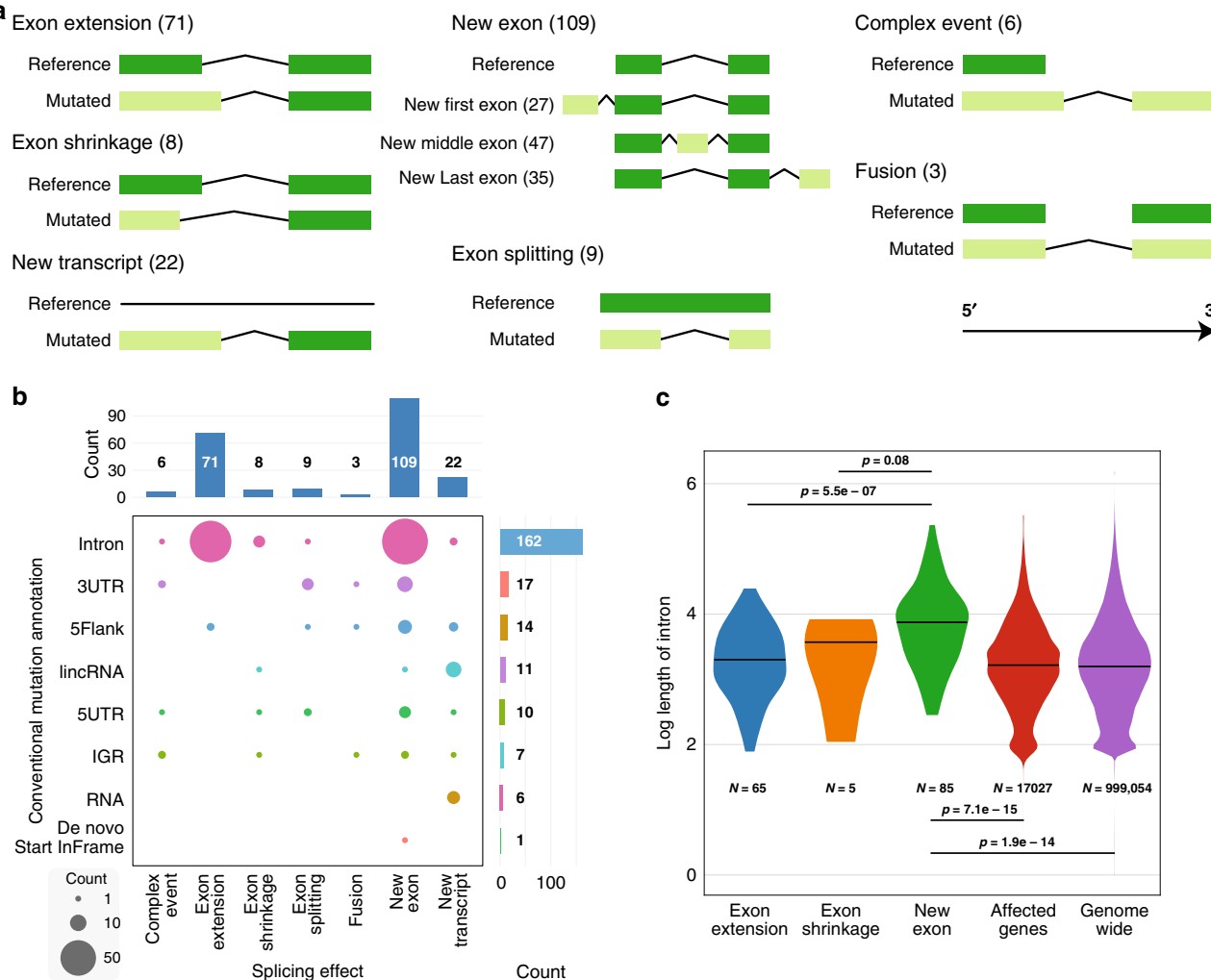

**Fig. 2 Category and distribution of nc-SCMs from TCGA WGS data. a** Schematics and counts of mutation-induced splicing events in seven different classifications. The new exon category is subdivided into first, middle, and last, depending on the location of the new exon. The complex event is a combination of multiple events, for example, exon extension and new exon. **b** Types and counts of splice forms in different non-coding regions. **c** Distribution of intron sizes (log 10 scale) across different categories. The p value is from the Wilcoxon's rank-sum test, two-sided. Source data are provided as a Source Data file.

canonical splice sites (Fig. 3a and Supplementary Data 1). Moreover, there are peaks for mutations near the splice site at −1 and +2 positions for donors and −1 position for acceptors (Fig. 3a).

Regarding creation of a new exon between two canonical exons, a mutation converts a nearby sequence to a functioning primary splice site (PSS), which in turn recruits an activated mate site (AMS) elsewhere in the intron (Fig. 3b). The PSS and AMS form the bounds of the new exon. AMSs have high splicing scores, averaging 5.13, which suggest that they were already primed for splicing but simply lacked a suitable splicing partner in the reference sequence (Fig. 3b, c). Supplementary Figure 3 shows consensus sequences in donor and acceptor splice sites for both the primary and activated mate sites. The percentages of T's in the −13 and −8 positions and G's in the −3 position all increase about 0.15 with the mutation in the acceptor splice site when compared to the reference. The percentages of A's and G's in the +3 and +5 positions in the donor splice site increase 0.3 and 0.15, respectively. This type of sequence motif resembles the consensus sequence in known donor and acceptor splice sites, suggesting the mechanism by which splicing scores increase. In addition, we found 33 events in which the new exons induce a frameshift, which leads to the truncation of the canonical protein

transcript in several genes, including five in cancer-related genes[21], namely, *ATRX*, *STK11*, *MAP4K3*, *MDM2*, and *MAX*. Four complex events involve concurrent exon shrinkage and the inclusion of a new exon, which occur in the UTR or IGR.

We annotated 228 splice-creating mutations with DNA variant allele fraction (VAF) and JAF values and their chromosome positions (Supplementary Fig. 4). We observe wide VAF and JAF variances, with all sites having values of at least 0.05 in both and averages of 0.36 and 0.37, respectively. There is no correlation between VAF and JAF values (Pearson's coefficient 0.028), with some genes, such as *ETV6* and *DNER*, showing low VAF values (0.08 and 0.15, respectively), but relatively high JAF values (0.33 and 1.0, respectively). Other genes, such as *MAP2K2* and *FOCAD*, have high VAF and JAF values, suggesting that the splice alteration is highly expressed and may have significant impact on gene function. The lack of correlation between VAF, which reflects sample purity, and JAF suggests that JAF is a reasonable indicator of splice junction strength. We also observed that some mutations affect more than one gene. For instance, we found one *RPL32-CAND2* fusion transcript caused by cryptic splice sites activated by a 3′-UTR mutation in *CAND2* (Supplementary Fig. 5). The 12875933C > G mutation at chromosome 3 disrupts

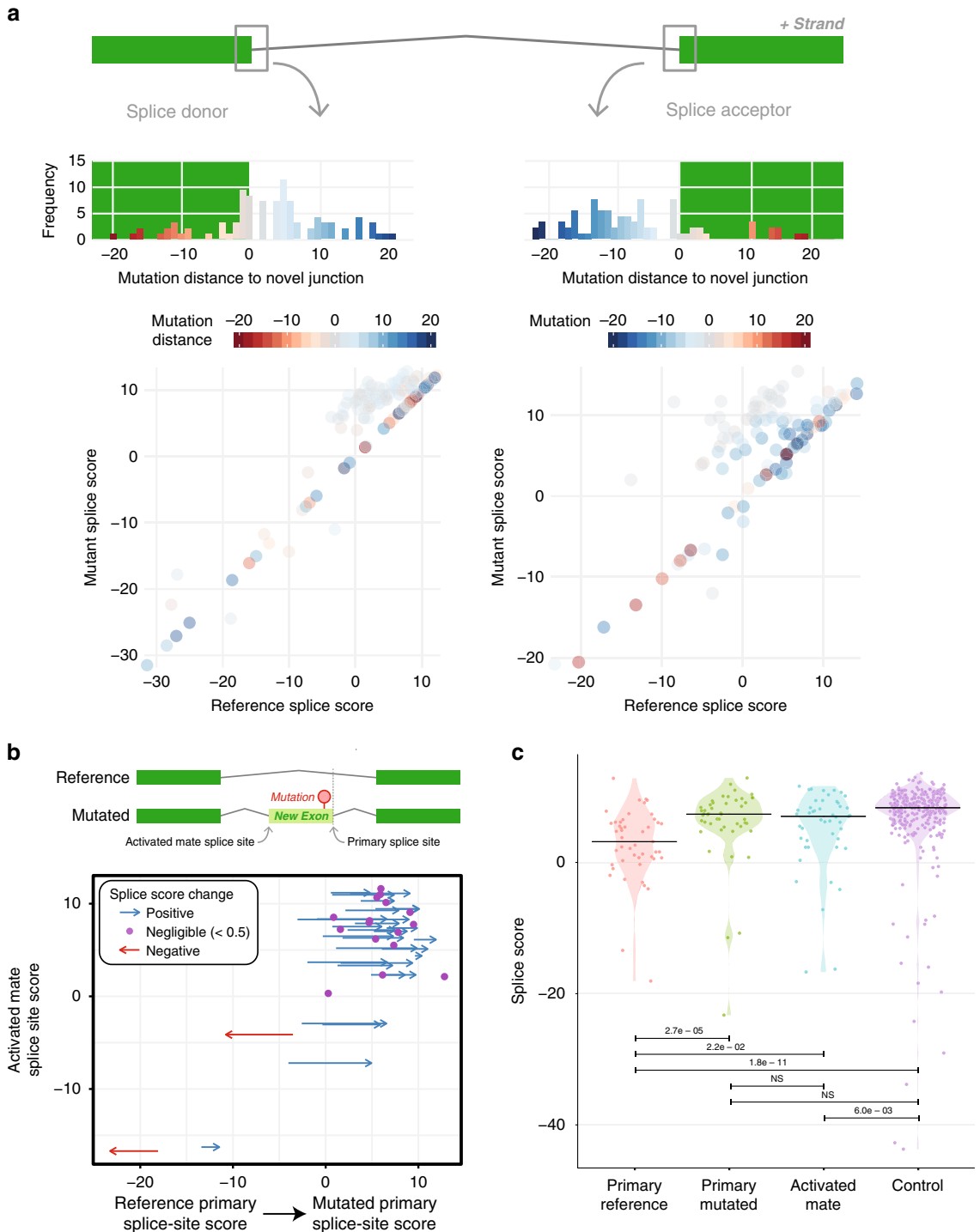

**Fig. 3 Characterization of nc-SCMs from TCGA WGS data. a** Frequency distribution of the distance from mutations to splice sites and splicing score comparison between the reference and mutant primary splice sites for donors and acceptors. Blue and red colors represent introns and exons, respectively. **b** Splice score distribution of new exon splice sites. Red arrows represent negative changes in score, blue arrows represent positive changes in score, and dots represent little to no changes in score (change <0.5). Arrow length is proportional to the magnitude of the change. **c** Comparison of primary splice score before and after mutation, the activated mate score in new exons, and the score of 300 random control sites. Source data are provided as a Source Data file.

the last coding exon of *RPL32* by skipping its stop codon and fusing the transcript with a partial 3′-UTR of *CAND2*.

**Effects of non-coding splice-creating mutations on protein sequences**. The 228 non-coding splice-creating mutations are found in 219 genes that are widely distributed across the genome

and of which 17 are associated with cancer, as reported by Lu et al.[21]. We observe a significant overrepresentation of nc-SCMs in cancer genes, including *NFE2L2*, *CDKN2B*, *FUBP1*, *MAP2K2*, *TBX3*, *FCGR3A*, *SETD2*, *DNER*, *ETV6*, *USP9X*, *MAP4K3*, *MDM2*, and *STK11* ($p < 0.001$); see "Methods." *CSPP1*, *STK11*, *SLCO1B1*, *NSMCE4A*, and *MAP4K3* contain two splice-creating mutations each, most of which have a DNA VAF and RNA JAF >20%.

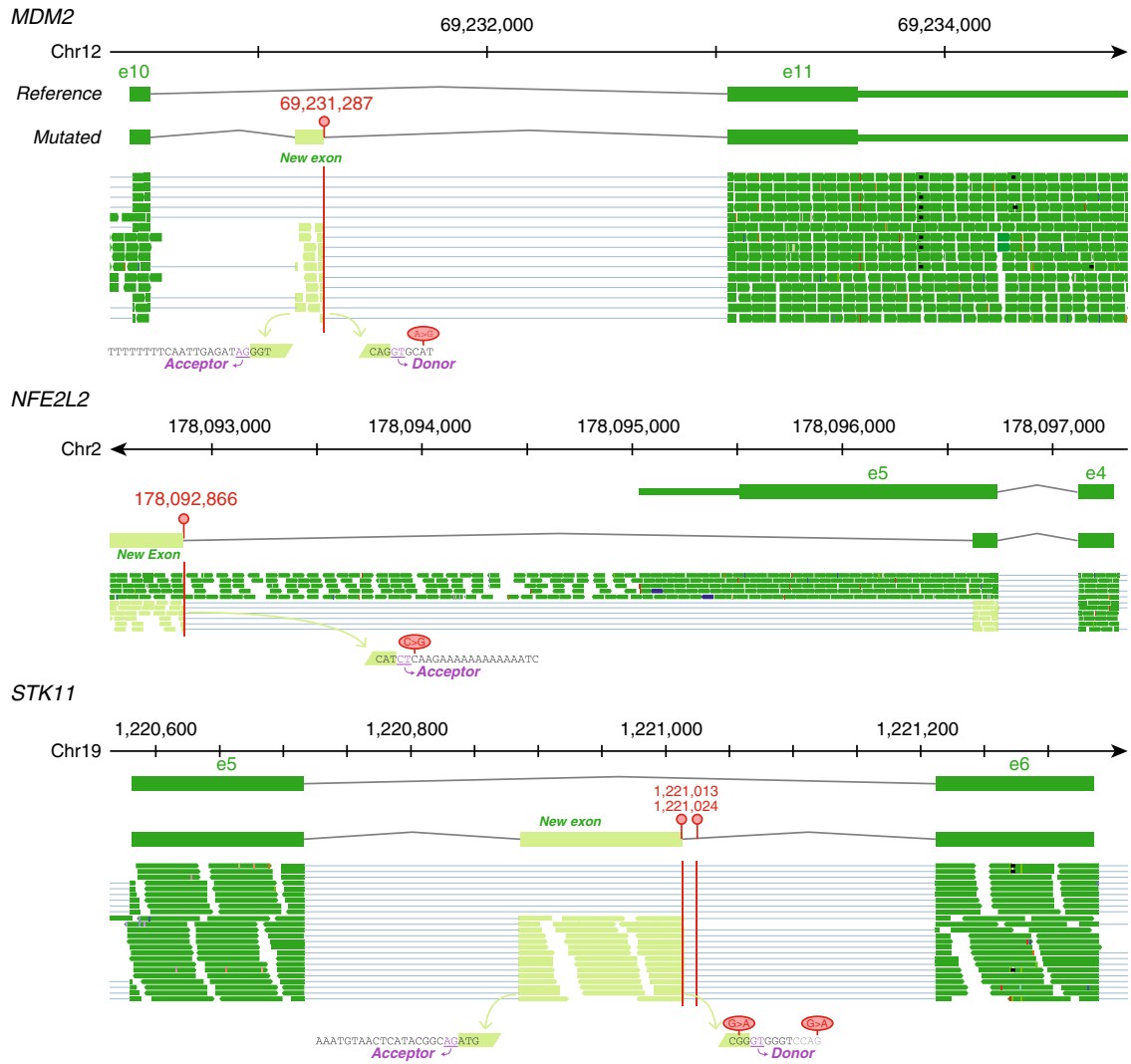

**Fig. 4 IGV schematics of *MDM2*, *NFE2L2*, and *STK11* splicing patterns.** Sequence reads visualized in IGV supporting the junctions of exon-creating events for *MDM2*, *NFE2L2*, and *STK11*. Red vertical lines indicate mutation positions. Source data are provided as a Source Data file.

For genes harboring non-coding mutations leading to alternative splice forms, we investigated the impact of the splice sites on gene expression, finding that 30 genes (>10%) with such sites are expression outliers ("Methods"), with 3 in the lower tail (*STK11*, *PPIL2*, and *HERC2*) and 27 in the upper tail (e.g., *SLCO1B1*, *PNPLA7*, *TFR2*, *TMEM10B*, *TNFRSF11A*, and *CAND2*) (Supplementary Fig. 6a). We note that these expression outliers did not pass the conventional false-discovery rate (FDR) <0.05 cut-off; see "Methods." The three events associated with low expression are frameshift or truncation events. We note that the ratio of expression outliers may be underestimated due to the limited control sample sets with both WGS and RNA-Seq data.

Notably, two proximal G > A intronic mutations in *STK11* from a head–neck cancer sample introduce a off-frame 130-bp exon between exons 5 and 6 (Fig. 4), which was also observed in previous study[13]. This development is associated with a lower gene expression of this tumor suppressor compared to the controls (Supplementary Fig. 6a). Loss of *STK11* function was found to be associated with the metastasis in various cancer types, such as lung and head–neck[22,23]. Importantly, we found that most of the splice-altering mutations we identified in intronic regions create new exons. For instance, a single A > G intronic mutation in *MDM2* from bladder cancer introduces a new 118-bp exon before the last canonical exon (Fig. 4). This new exon carries

a stop codon, leading to the truncation of the protein transcript and the loss of a RING-type zinc-finger domain. The p53-suppressive activity of wild-type (WT) MDM2 can be inhibited by the ribosomal protein L11[24]. *MDM2* zinc-finger mutants, however, can escape inhibition to play important roles in tumor progression[23]. In another example, an intergenic C > G mutation 2166 bp from the 3′-UTR of *NFE2L2* in a lung cancer case leads to a new junction that pairs with a site embedded in the start of the last exon of *NFE2L2* (Fig. 4), potentially disrupting NFE2L2/KEAP1 interaction. Dysfunctional NFE2L2 and KEAP1 interaction was found to be an activating factor for NFE2L2 oncogenic function in lung cancer[25]. This splice alteration results in the loss of most of the transcript of the last exon and the mutant transcript has >10 RNA-supporting reads for the junction (Supplementary Data 1). These examples illustrate potential oncogenic effects of these non-coding mutations through altered splicing.

**Experimental validation of non-coding splice-creating mutations from 9494 TCGA exome data.** The wingspan of exome probes from TCGA contains non-coding regions, which are close to exon boundary such as 5′- and 3′-UTRs and introns. To further investigate the dynamics of nc-SCMs, we performed

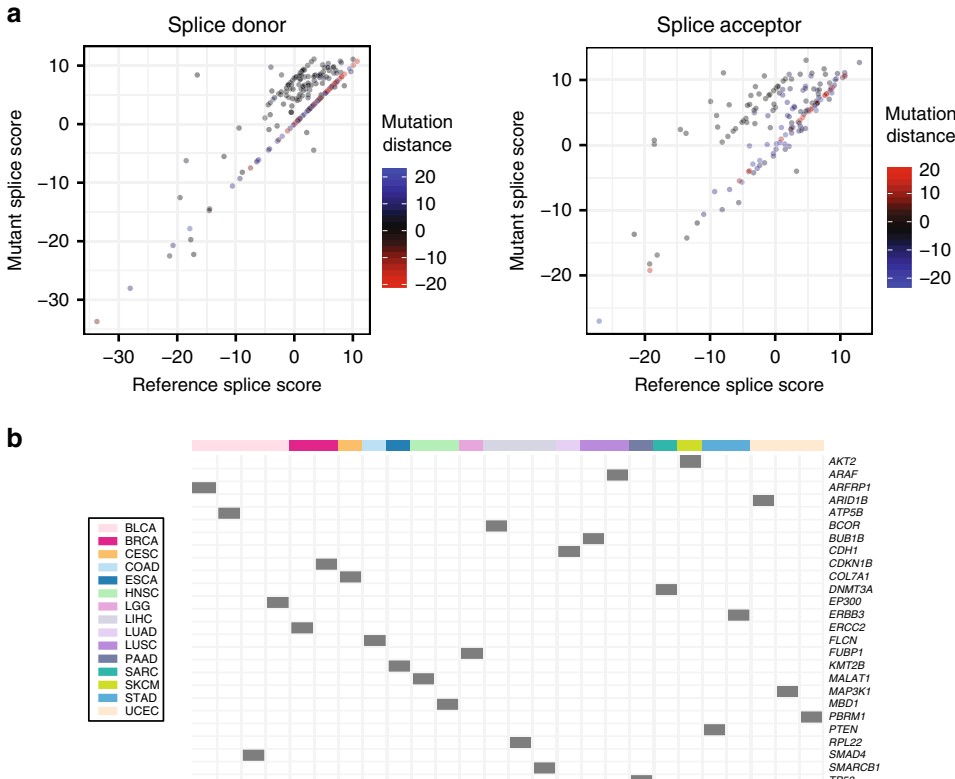

**Fig. 5 Characterization of nc-SCMs from TCGA WES data. a** Comparison of splice scores between the reference and mutant for donors and acceptors. The color indicates the mutation distance to the junction. **b** nc-SCMs found in cancer-related genes across different cancer types. Source data are provided as a Source Data file.

additional large-scale discovery from a much more substantial cohort, 9494 TCGA tumor exomes across 33 cancer types. We extracted non-coding mutations from the unfiltered MC3 MAF file and ran MiSplice on these mutations ("Methods"). We found 389 nc-SCMs, of which 369 passed manual review with a 5% false-positive rate (Supplementary Data 2). Figure 5a shows the splice scores before and after mutations for alternative splice site (AltSS). In general, SCMs result in an increase of splice score of AltSSs: 57% of SCMs increase the splice score of AltSS, 33% of SCMs show comparable splice score, and only 10% of SCMs shows a reverse trend.

Supplementary Figure 6b shows 57 genes with nc-SCMs, which are expression outliers, 9 and 48 of which are in low and high tails, respectively. Supplementary Figure 6b shows 26 nc-SCMs residing in cancer genes across 15 cancer types, namely, *AKT2*, *ARAF*, *ARFRP1*, *ARID1B*, *ATP5B*, *BCOR*, *BUB1B*, *CDH1*, *CDKN1B*, *COL7A1*, *DNMT3A*, *EP300*, *ERBB3*, *ERCC2*, *FLCN*, *FUBP1*, *KMT2B*, *MALAT1*, *MAP3K1*, *MBD1*, *PBRM1*, *PTEN*, *RPL22*, *SMAD4*, *SMARCNB1*, and *TP53*. We note that these expression outliers did not pass the FDR <0.05 cut-off; see "Methods."

We used an established mini-gene assay ("Methods") to experimentally examine the splicing alterations generated by five selected nc-SCMs in cancer-related genes[21], namely, *EP300*, *BCOR*, *DNMT3A*, *KMT2B*, and *MAP3K1* (Fig. 6a), which play important roles in cancer initiation and progression. The additional band in the mutant product at ~235 bp represents endogenous exons in the pCAS2.1 plasmid without the mutant exon. As shown in Fig. 6a, often with this assay, if the alternative mutant splice site is not strong enough, we see multiple alternatively spliced products for some of the mutant constructs. This is an expected observation with the pCAS2.1 plasmid and the mini-gene splicing assay, since the endogenous exons have

very strong splice sites[16]. Overall, we validated four splicing alterations, DMNT3A being the exception, for a suggested 80% validation rate. The results of Sanger sequencing confirmation of reverse transcription-polymerase chain reaction (RT-PCR) of *EP300*, *BCOR*, *KMT2B*, and *MAP3K1* mutants can be found in Supplementary Figs. 7–10, which support the extended/shrunken exons created by intronic mutations. Taken together with a previous study of ours on coding SCMs that showed high validation rate, the mini-gene result is mostly consistent with the predicted result of the MiSplice pipeline for both coding and non-coding SCMs. Figure 6b shows the IGV diagrams of alternative splicing products resulting from intronic mutations in *EP300*, *KMT2B*, *MAP3K1*, and *BCOR*. The intronic mutations of *EP300* and *KMT2B*, which are 10 and 8 bp away from the nearest exon, respectively, create cryptic AG splice sites due to the T -> G mutations. The result is exon extension and generation of in-frame and frameshifting products, respectively. In addition, Fig. 6b shows that intronic mutations close to the acceptor AG and donor GT sites weaken the canonical splice sites and result in the AltSSs, which produce a 29-bp exon extension in *MAP3K1* and a 9-bp exon shrinkage in *BCOR*.

## Discussion

Here, we applied MiSplice to detect non-coding splice-site creating mutation. Benchmark tests on simulated data show MiSplice has relatively high sensitivity (>0.74) when given coverages of >20× for junctions, with manual review showing comparable specificity of about 0.77. Application of the pipeline to TCGA WGS and TCGA WES mutation data and RNA-Seq data highlighted 562 mutation-induced splicing alterations in non-coding regions of the genome. Analysis shows that mutations increase the splicing score in a splicing region by either creating

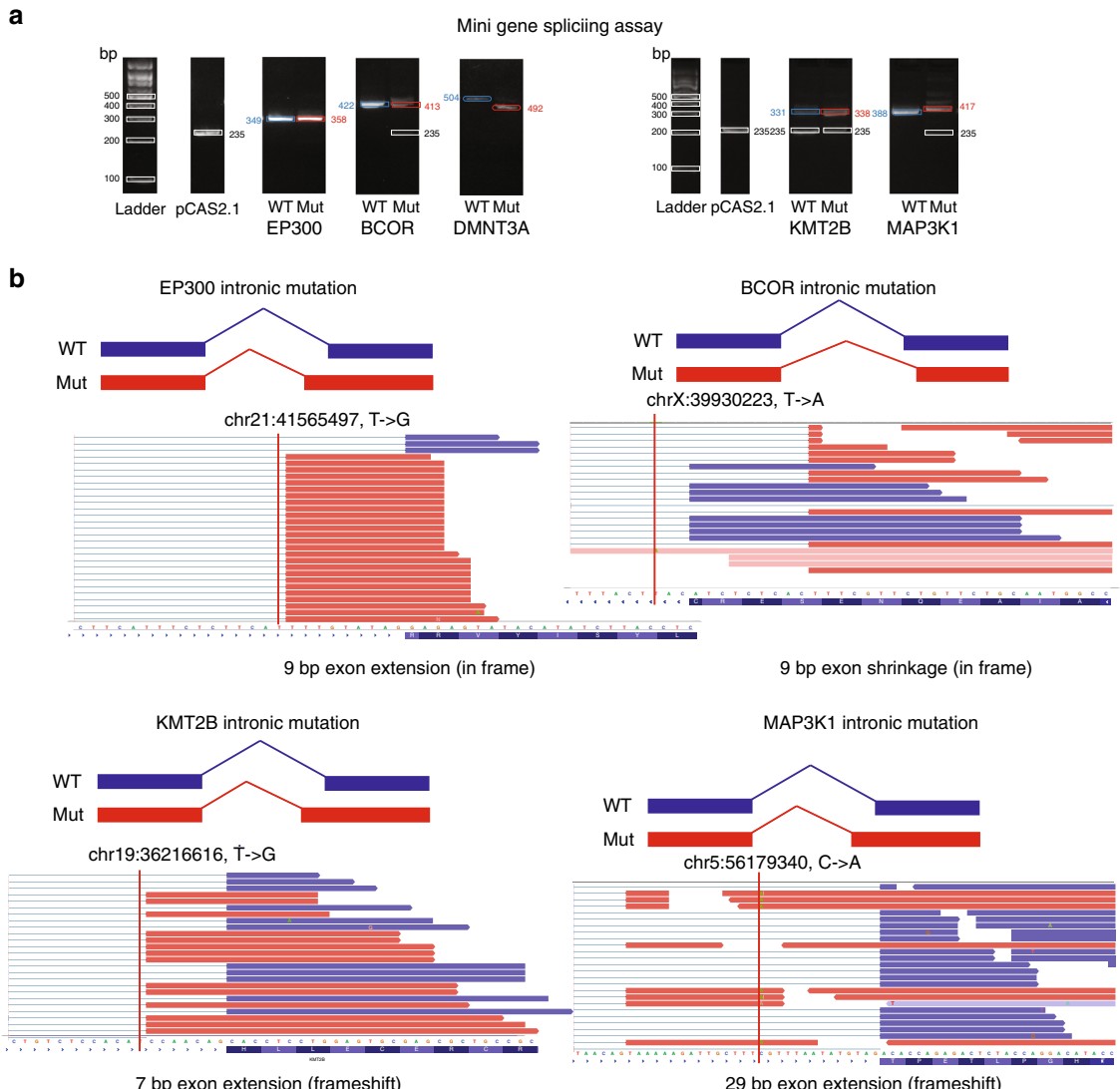

**Fig. 6 Experimental validation of nc-SCMs by the mini-gene splicing assay. a** Validation of non-coding splice-site-creating mutation candidates by the mini-gene splicing assay ("Methods"). All experiments were performed by three replicates and gave similar results (see Source Data file for the full scans). **b** Integrative Genomics Viewer (IGV) screenshot of four validated mutation-induced exon extension or shrinkage events. Red vertical lines indicate the mutation positions. Source data are provided as a Source Data file.

alternative canonical splice sites or favorably altering splicing sequence context. We also found that new exon creation is the dominant splicing alteration in non-coding regions and that corresponding introns have a larger size compared to alternative splicing events. In addition, we observed more cases with high expression outliers for genes harboring nc-SCMs in both WGS and WES data, which may reflect a decay mechanism of the altered splice product[26,27].

A notable aspect of these findings is that non-coding splice-altering mutations lie in several cancer-associated genes, such as *ATRX*, *BCOR*, *CDKN2B*, *MAP3K1*, *MAP3K4*, *MDM2*, *SMAD4*, *STK11*, and *TP53*. For instance, two proximal G > A intronic mutations result in a new 130-bp exon in *STK11*, which is associated with lower gene expression. *STK11* (or *LKB1*) is a tumor suppressor gene and its loss of function is associated with tumor metastasis in lung and head–neck cancers[22,23]. These findings suggest adding analyses of splicing altering potentials in non-coding mutations to cancer studies to ensure that these mutations are not overlooked based on their conventional annotations.

In conclusion, our study demonstrates that 16% of tumors from various cancer types harbor non-coding mutations that create alternative splice sites based on TCGA WGS data, which is a higher figure than the 1% and 9% of tumors with nc-SCMs from our 2018 study[16] and the Shiraishi's work[18], respectively. These findings shed further light on the oncogenic effects of splice-creating non-coding mutations. Applying this approach more broadly will help unveil critical non-coding mutations associated with tumorigenesis in cancer. We anticipate that larger sample sizes and inclusion of additional cancer types with WGS data will lead to the discovery of more splice-creating mutations. While recurrence is often exploited to increase statistical power, there are only two recurrent nc-SCMs in the current study. Since lack of recurrence seems to be a general feature of our data, we did not consider it in the current study. The bioinformatics problem of finding and analyzing non-coding events and identifying subsets of contributing drivers still poses a great challenge in the field of cancer genomics. Our observations, coupled with the fact that the data corpus of human sequences continues its rapid expansion, especially as multi-omics data are starting to be generated in the

clinical setting, point to the urgent need for purpose-built bioinformatics tools that can manage the analysis of non-coding mutations and their impact on human health and disease.

## Methods

**MiSplice pipeline**. The MiSplice pipeline was developed to detect mutation-induced splicing events from RNA-Seq data. It is written in Perl and uses two standard tools, SAMtools and MaxEntScan. The pipeline is fully automated and can run multiple jobs in parallel on a compute cluster. It consists of the following controller, processing, and helper modules:

- *Top-level controller module—orchestrating execution*: A maf file is split into multiple smaller files, each containing a subset of mutations (the default setting of 200 was used here) in order to process these in parallel on a compute cluster. The top-level routine handles the parceling task, LSF job queuing, and management of individual processing steps below.
- *Processing module 1—junction discovery*: A search is executed for splice junctions within $N = 20$ bp of each subject mutation having at least $M = 5$ supporting reads (see minimum information power calculation below), each with mapping quality $Q \geq 20$. Canonical junctions are then filtered using the Ensembl 37.75 database. We selected 20 bp as a cut-off since it is the farthest distance from the splice junction in a splice region. A comparable calculation for the number of supporting reads is made for the above cryptic splice sites in control samples without mutations and this information is later passed to processing module 4.
- *Processing module 2—known junctions/polymorphic genes*: Cryptic sites that fall within polymorphic gene loci, for example, *HLA*, or those proximal to known junctions are removed.
- *Processing module 3—coverage/JAF*: Sites failing either a heuristic ranking threshold, that is, having <5% of reads supporting the junction, or (optionally) sites failing a proportion test of number of reads at the genomic location supporting the junction of interest and FDR correction are filtered out.
- *Processing module 4—case/control*: Further filtering of cryptic sites is done by comparing the supporting reads in control samples. The final reported cryptic sites must stand as the top $k = 5\%$ for the number of supporting reads in the case (with mutation).
- *Helper module—scoring*: Splicing scores for the cryptic splice sites are calculated using MaxEntScan.
- *Helper module—read counting*: Read depth of each cryptic splice site is calculated and reported by SAMtools.

**Simulation for estimating sensitivity**. Because there is not yet a ground truth set for the type of splice alterations examined here, sensitivity is more difficult to estimate than specificity, the latter being a by-product of the manual review process. We sought to estimate the sensitivity via constructing a simulated truth set. Genomic sequences of length $10^6$ bp were generated in silico in FASTA format with an expected GC (guanine–cytosine) fraction of 40%. Sequences for each gene were simulated with 100 exons, with the lengths of each exon and intron being picked randomly within the ranges of 15–600 and 100–1000, respectively. Transcripts were formed by appending exon sequences in that gene, adding 3′- and 5′-UTRs and a poly(A) tail. Exon extensions were simulated by extending the exon length to 5′ or 3′ ends by a defined number of base pairs ranging from 5 to 50 bp. Sequence reads of 75 bp were then simulated by picking random positions within the transcript, with nucleotides recapitulating the reference sequence at a rate specified by $Q = 75$ base quality, and substitution rate of 0.001. Simulation of deletions and insertions was omitted, as these are not primarily characteristic of Illumina data. Mixed simulated data were generated by combining canonical exon transcripts with exon extension transcripts at fractions ranging from 0.1 to 0.5.

**Proportion test**. An alternative to the heuristic ranking is using a binomial proportion test by estimating the Bernoulli probability that any arbitrarily selected read would support a site. This value can be used to filter sites based on whether the number of supporting reads for a given site is significantly higher than expected by chance. Standard Benjamini–Hochberg FDR multiple test correction can subsequently be applied across the sites.

**Expression outlier analysis**. To investigate if genes harboring splice sites are expression outliers, we used Tukey's standard formula to quantify an outlier score:

$$\text{Outlier score} = (x - Q3)/\text{IQR for upper tail and}(x - Q1)/\text{IQR for low tail}, \quad (1)$$

where IQR is the interquartile range, Q1 and Q3 are the first and third quartiles, respectively, and $x$ is the RSEM (RNA-Seq by expectation maximization) value in a log 2 scale. In the current study, genes with an outlier score >1.5 or <−1.5 are considered to be expression outliers. We converted outlier score to $p$ value and did the Benjamini–Hochberg FDR multiple test correction by using R package.

**Fisher's exact test**. We calculated $p$ values for the overrepresentation of nc-SCMs in cancer genes by Fisher's exact test. Based on Ensemble 75 data, there are about 20,000 genes in the human genome, of which 624 are cancer-related genes[21]. We found that 228 nc-SCMs are widely distributed in 219 genes, roughly each gene having a unique nc-SCM, of which 17 are cancer-related genes. We created the following 2 × 2 table:
$\begin{bmatrix} n1 & n2 \\ t1 & t2 \end{bmatrix} = \begin{bmatrix} 202 & 17 \\ 19,376 & 624 \end{bmatrix}$, where $n1$ and $n2$ are the numbers of non-cancer and cancer-related genes with nc-SCMs, and $t1$ and $t2$ are respective tallies from the overall genome. The table yields a two-sided Fisher tailed $p$ value is <0.01.

**Power assessments**. A primary issue in all detection algorithms is the minimum information required to be able to discern a real signal. In keeping with the binomial approach of assessing numbers of supporting reads versus WT reads for a candidate site, we estimate required read depth using the standard Gaussian-approximated power formula for one-sided binomial testing.

$$R \geq \left( \frac{Z_{1-\alpha}\sqrt{J_0(1-J_0)} + Z_{1-\beta}\sqrt{J_1(1-J_1)}}{J_1 - J_0} \right)^2, \quad (2)$$

where $R$ is the required number of covering reads, $Z_{1-\alpha}$ and $Z_{1-\beta}$ are the respective type I (false-positive) and type II (power) Z-scores, and $J_0$ and $J_1$ are the null and alternative JAFs. It is common to select type I and II Z values of 1.645 and 0.84, respectively, representing 5% false positive and 80% power. Figure 1b was plotted assuming a null JAF of 5%. The generally deep coverage can result in higher tendencies for artifactual alignments, which can be misinterpreted as supporting evidence. For 200× depth, the expected number of artifacts is $200 \times 10^{-Q/10} = 2$ reads, assuming average quality of $Q = 20$. We therefore added a minimum requirement of five junction supporting reads based on the Poisson tail probability that having such a combination of artifacts purely by chance would be only ~5%.

**Long intron bias estimation**. New exon creation depends upon finding both a 9-nucleotide donor and a 23-nucleotide acceptor, a joint event of 32 nucleotides having a Bernoulli probability of $1/4^{32} \approx 10^{-19}$. In addition, within an intron of length $L$, the donor and acceptor must be separated from each other by a distance $E$ that is characteristic of exon size, which can occur in roughly $L - E - 31$ ways. If we omit consideration of the mutation itself, the probability of realizing the conditions for new exon creation is then proportional to $1 - (1 - 10^{-19})^{L-E-31} \approx 1 - \exp[-10^{-19}(L - E - 31)] \approx 10^{-19}(L - E - 31)$. The last simplification stems from expanding $e^{-x}$, where $x$ is small and only the first term is retained. It implies that bias of long introns versus shorter introns, $L_L$ and $L_S$, respectively, can be quantified by the ratio $(L_L - E - 31)/(L_S - E - 31)$. For the types of smaller introns found here, this ratio is approximately $L_L/L_S \approx 30,000/3000$, indicating a roughly 10-fold bias toward large introns in the data we examined. However, this model also implies an additional bias in the form of an "edge effect" against larger exons arising within smaller introns, for example, for $E = 200$ and $L_S = 1000$ the ratio is ~40-fold. Such cases would be expected only in rare instances.

**Splice site score estimation**. For each cryptic splice site and nearby canonical splice site, the corresponding nucleotide sequences were first extracted for both the mutant and reference sequences (9 mer and 23 mer for donor and acceptor sites, respectively). Their splice scores as potential donor or acceptor sites were then estimated using MaxEntScan with a maximum entropy scoring model.

**IGV manual review**. As described above, the MiSplice pipeline uses control and case comparison to remove cryptic sites, which were found in samples without mutations. In the manual review step, we further looked into the mutations at the RNA level by IGV to further remove false positives. We specifically double-checked that at least five reads support the junction. Also, for spliced-in mutations, which are included in the newly created exon from the cryptic site (see the MAP3K1 chr5:56179340 intronic mutation in Fig. 6b), we double-checked whether reads, which cover the cryptic site and mutation site, contain the specific mutation. If not, we assigned it as a false positive and removed it from the downstream analysis.

**Mini-gene splicing assay**. Exons and flanking sequences from HEK293T genomic DNA are amplified using primers carrying restriction enzyme sites for *Bam*H1 and *Mlu*I. Amplified sequences are subject to NucleoSpin PCR Cleanup (Macherey-Nagel) or DNA Clean and Concentrator-5 Kit (Zymo Research) and digested with *Bam*HI and *Mlu*I. T4 DNA ligase (NEB) was used to ligate the digested pCAS2.1 vector and amplified sequence. The RT-PCR sequence of pCAS2.1 is "TGACGT CGCCGCCCATCACGCCTCCAGGCTGACCCTGCTGACCCTCCTGCTGCT GCTGCTGGCTGG**GG**ATAGAGCCTCCTCAAATCCAAATGCTACCAGCTCC AGCAGCCAAGAT**CC**AGAGAGTTTGCAAGACAGAGGCGAAGGGAAGGTC GCAACAACAGTTATCTCCAAGATGCTATTCGTTGAACCCATCCTGGAGG TTTCCAGCTTGCCGAC AACCAACTCAACAACCAAT." The two bold and underlined "**GG**" nucleotides are the boundary between the two endogenous exons in the pCAS2.1 plasmid, which is the inserted position of the exon of WT or mutant. Mutations were introduced via Q5 Site-Directed Mutagenesis (NEB). Both the inserted amplified sequence and the mutation of interest were confirmed by

sequencing. Plasmids are transiently transfected with Lipofectamine 2000 (Thermo Fisher Scientific) into HEK293T cells. After 24 h post transfection, cDNA was synthesized using 2–3 μg of total RNA with the Superscript III First-Strand Synthesis System (Thermo Fisher Scientific) and Oligo(dT)20 was used for priming. cDNA amplification was performed with the following primers: pCAS-KO1- (5′-TGACGTCGCCGCCCAT-3′) and pCAS-R (5′-ATTGGTTGTTGAGTTGGT TGTCGG-3′). Splicing patterns were visualized on a 4% agarose gel with ethidium bromide. Finally, each alternative band on the gel was purified for sequencing using Qiaquick Gel Extraction Kit (Qiagen) to validate mutant and WT predictions[16]. All primer sequences used in this study can be found in Supplementary Table 1.

**Reporting summary**. Further information on research design is available in the Nature Research Reporting Summary linked to this article.

## Data availability

The WGS mutation data for 790 TCGA samples with RNA-Seq were obtained from the International Cancer Genome Consortium (ICGC) at https://www.synapse.org/#! Synapse:syn7118450 (version 12-Oct-2016). We removed one outlier cancer type (DLBC) with only seven samples, which reduced the WGS samples set from 790 to 783; see Supplementary Fig. 2. The full name of each cancer type included in the current study can be found at https://gdc.cancer.gov/resources-tcga-users/tcga-code-tables/tcga-study-abbreviations. The controlled-accessed WES mutation data were downloaded from GDC (https://gdc.cancer.gov/about-data/publications/mc3-2017). The ISB-CGC (https://isb-cgc.appspot.com) access of the TCGA RNA-Seq bam corpus was granted through tcga-phs000178-controlled credential. The TCGA RNA-Seq alignments used in this study were generated by using MapSplice (https://academic.oup.com/nar/article/38/18/e178/1068935) against the hg19 reference genome. Details needed to replicate TCGA RNA-Seq bam file can be found at https://webshare.bioinf.unc.edu/public/mRNAseq_TCGA/. We also obtained gene expression data (RSEM) from the Broad firehose collection (http://gdac.broadinstitute.org/runs/stddata__2016_01_28/) across 33 TCGA cancer types. Ensembl 37.75 database can be downloaded from ftp://ftp.ensembl.org/pub/release-75/gtf/homo_sapiens/. All other data supporting the findings of this study are available from the corresponding author upon request. Source data are provided with this paper.

## Code availability

MiSplice is written in Perl and is freely available at https://github.com/ding-lab/Misplice under the GNU general public license. MiSplice uses several independent tools and packages, including SAMtools and MaxEntScan, all of which are likewise freely available, but which must be obtained independently from their respective developers. The MiSplice documentation contains complete instructions for obtaining and linking these applications into the MiSplice pipeline.

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

## Acknowledgements

This work was supported by the National Cancer Institute grants R01CA178383, R01CA180006 and U24CA211006 to L.D. F.C. is supported by National Institute of Diabetes and Digestive and Kidney Diseases grant R01DK087960. Additional support came from the National Institute of General Medical Sciences Cell and Molecular Biology training grant GM 007067 (R.G.J.). The Cancer Genome Atlas (cancergenome.nih.gov) and The International Cancer Genome Consortium (ICGC) were the source of primary data. We acknowledge support of computational resources from McDonnell Genome Institute, the Oncology Division of the Washington University School of Medicine, and the Institute for Systems Biology-Cancer Genomics Cloud (ISB-CGC), a pilot project of the National Cancer Institute (under contract number HHSN261201400007C).

## Author contributions

L.D. designed and supervised research. F.C. supervised the experimental design and the biological evaluations. S.C., D.C.Z., C.J.Y., M.H.B., T.T., Q.G., S.R., and I.S. analyzed the data. C.J.Y. generated the in silico reads for the simulations. C.O., R.G.J., Y.Z., and A.M. conducted splicing experiments. S.C., D.C.Z., M.A.W., and M.H.B. prepared figures and tables. S.C., D.C.Z., M.C.W., F.C., and L.D. wrote the manuscript. F.C., M.C.W., and L.D. revised the manuscript. We thank Steven Foltz, Sam Sun, and Kuan-lin Huang for critical reading of the manuscript.

## Competing interests

The authors declare no competing interests.
