## [Peer Review File · Nature Communications]

Reviewers' comments:

Reviewer #1 (Remarks to the Author): Expertise in bioinformatics, transcriptomics and splicing

Cao et al. use whole-genome (WGS), whole-exome (WES), and matched RNA sequencing data from TCGA to identify non-coding mutations that create new splice sites. This appears to be an extension of their previous work, Jayasinghe et al. Cell Reports 2018. Jayasinghe et al. describes their bioinformatic tool MiSplice and its application to 8,656 TCGA tumors from TCGA, but only based on exome sequencing. Here, they extend the analysis to 9,494 TCGA samples and 790 whole-genome samples. The addition of whole-genome sequencing data, in particular, allowed them to identify novel exon creation in key cancer genes. Although this represents a more comprehensive analysis due to the joint analysis of WGS and WES, it is unclear how many of the non-coding splice-site-creating mutations (nc-SCMs) are unique to this analysis when compared to their own publication (Jayasinghe et al. Cell Reports 2018), results from WES/RNA-Seq analysis with the SAVnet approach (Shiraishi et al. Genome Research 2018), or WGS/RNA-seq analysis from the PCAWG consortium (PCAWG Transcriptome Core Group bioRxiv 2018). All three of these studies have included non-coding (intronic) mutations that create splice sites. The authors should perform a comparison of the results as well as comment on differences in methodologies that make this study more unique than these previous studies.

In addition, here are major and minor comments/suggestions:

MAJOR

1. P. 5: When filtering their identified nc-SCMs, they reduced false positives by filtering splicing events observed in at least 10 control samples. How did they determine the threshold of 10 control samples?
2. P.5: They also state that "sites with spliced-in mutations were required to have at least 30% supporting reads...". It is unclear what are "sites with spliced-in mutations". Does that mean the mutation was observed in the RNA-Seq read? What is the basis of this specific filter?
3. Exon creation is described in the PCAWG study and should be referenced.
4. P.6, starting at line 167, the authors characterize new exons that "often" carry a stop codon or that "most" complex events involve concurrent exon shrinkage. Exactly how many of these are there?
5. The authors highlight a CAND2-RPL32 fusion transcript that is associated with a splice site creating mutation. Should this be written as RPL32-CAND2? Based on the Supp Figure and that the mutation creates a 3' splice site, I would assume that fusion transcript would have RPL32 at the beginning, followed by CAND2.
6. Line 199 states that 28 genes had expression outliers, but Figure 5 shows 30 and the legend also states there are 30.
7. Regarding the mini-gene assays, it is unclear why the particular 5 genes were selected for validation. Also, for some of the mutant products shown in their gels on Fig. 7A, it is very difficult to resolve a difference in products size. For example, the EP300 mutant product is 9bp different from the wild-type product. The gels shown cannot resolve this difference. They look like the same size. Moreover, I'm unclear what is exactly shown in the Sanger sequencing validation shown in Supplementary figures. The legends say it highlights the plasmid boundary. Wouldn't you want to sequence the mutant junction product? And if so, this would be difficult to do given the poor resolution on the gel.
8. In their discussion, they should expand a bit on their observation that they see more cases of increased outlier expression of the genes rather than a decrease in outlier expression. This perhaps mean that there are additional nc-SCMs that are missed because the splice product is degraded and cannot be detected by observing the splice junction.
9. Line 283: They claim this is the first study to demonstrate that 20% of tumors harbor mutations that disrupt splicing patterns. Again, how does this compare to their own 2018 study and the Shiraishi study?
10. Figure 2C: I'm not sure what the black line is supposed to indicate. The black line on the violin plot for the "New exon" is at a location I wouldn't expect. Usually these are at the median value, but it doesn't appear to be on the plot. Also, there is a letter "A" typo over the "affected genes" violin plot. Perhaps this is a PDF conversion issue.

11. Figure 7B: It is unclear what the red line indicates? It seems to be placed on the created splice site for EP300 and KMT2B, but for the other two genes, the red line seems to be at a random location.

MINOR

1. Instances of "splicing site" should be changed to "splice site":

a. In the title

b. Also subheader on p.8 should be changed from "non-coding splicing-creating" to "splice-creating".

c. Line 236: "alternative splicing sites" changed to "alternative splice sites"

2. P.3, last sentence: "alternative splicing phenomena enriched to non-coding regions" should be changed to "alternative splicing phenomena enriched in non-coding regions"

3. P. 4, line 86: Remove the word "respectively"

4. line 94: There is an author's comment that should be removed

5. line 96: "perform" should be changed to "performance"

6. line 117: "no-coding" should be changed to "non-coding"

7. line 268 mentions this is ICGC WGS. Do they mean TCGA WGS?

8. I find Figure 4 to be somewhat uninformative. If there is a need to reduce the number of figures, I would suggest this be moved to supplement.

Reviewer #2 (Remarks to the Author): Expertise in bioinformatics, transcriptomics and splicing

Review comments:

The authors use available WGS, WXS and RNAseq data to identify somatic mutations which potentially cause splicing alterations in cis. Their method aims at discovering a high confidence set of 'novel' junctions near (~20nt) splice site creating mutations, integrating various parameters like supporting reads per mutation, junction, mapping quality and filtering against canonical junctions. The authors find a reasonable sized set of candidate mutations that track well with observed splicing alterations and then aim to characterize general impact patterns these mutations have on splicing. Furthermore, the authors investigate transcript changes in some target genes of interest (in this case cancer related genes). Overall, this manuscript shows the utility of integrating WGS and RNAseq data to stratify somatic mutations in non-coding regions and gives some insight how even non-coding mutations can be functionally relevant in disease by altering splicing patterns.

Major points:

- How exactly were the 281 splice altering mutations manually reviewed in IGV? What was assessed and what criteria were used? This needs at least some section in methods and clarification/justification why there was a need for manual review.

- Fig3A enrichment in mutations +/- 3 bp from SS, the data presented do not show that. Donors have enrichments near the SS in both intron and exon, whereas acceptors have a peak at -1 (but not as the authors claim -3).

o Mutations that destroy the canonical AG and GU motif have already been annotated (e.g. ensemble's variant effect predictor) – the authors should address how many of their non-coding variants are actually altering splicing beyond destruction of canonical splice sites

- The authors state: "17 out of 219 genes are associated with cancer" and later that this 'constitutes a significant enrichment in cancer genes affected by nc mutations.' How is that enrichment analysis performed? What is the reference gene set for cancer associated genes? COSMIC? What is the enrichment being compared to? Random gene sets? This needs further explanation in the method section. Given the extensive size of cancer associated genes, any random gene list of >200 genes will likely feature 10-20 cancer associated genes. Is this enrichment or just expectation?

- What are the JAVs and gene expression changes for cancer related genes? Why no percentage spliced in (PSI)?

- Also, regarding expression outliers there might be some need for multiple testing correction, given only a 5% FDR for 369 mutations spread over 33 tumor types, many reported expression

outliers might be coincidence.

- MiSplice tool is already published and no new methodological changes are made (as is also admitted in the text - see lines 94-96).
- language sometimes quite vague (e.g., line 102: reasonably sensitivity)
- Power calculation is per sample. Have the authors thought about combining evidence from multiple samples to look for lower-coverage but recurrent changes, possibly increasing power.
- The minimal read depths shown in the power calculation of Figure 1B is 10, but the actual coverage threshold used for discover is 5 (line 122). Why?
- For the outlier analysis it is not quite clear what the distributions are over. For instance in Figure 1B - STK11 is under-expressed in HNSC. Does this only occur in HNSC? Is the distribution over all HNSC samples?

Minor points:

- ISB-CGC in line 301 misses a version / access time (how were the RNA-Seq alignments generated?)
- line 356: Is the correction applied across all sites or what is referred to by trials?
- Figure 2C: Do the asterisks refer to different significance levels?
- Image quality of minigene reporter agarose gels
- o Why are there varied size PCR amplicons in MUT but not WT PCR for MAPK, KMT2B and BCOR?

Typos:

- * Figure 2C: axis label: Log lenth  Log length
- * Figure 3A: axis label: Spice Site  Splice Site

Responses to Reviewers

Reviewer #1 (Remarks to the Author)

The addition of whole-genome sequencing data, in particular, allowed them to identify novel exon creation in key cancer genes. Although this represents a more comprehensive analysis due to the joint analysis of WGS and WES, it is unclear how many of the non-coding splice-site-creating mutations (nc-SCMs) are unique to this analysis when compared to their own publication (Jayasinghe et al. Cell Reports 2018), results from WES/RNA-Seq analysis with the SAVnet approach (Shiraishi et al. Genome Research 2018), or WGS/RNA-seq analysis from the PCAWG consortium (PCAWG Transcriptome Core Group bioRxiv 2018). All three of these studies have included non-coding (intronic) mutations that create splice sites. The authors should perform a comparison of the results as well as comment on differences in methodologies that make this study more unique than these previous studies.

We thank the reviewer for the positive assessment of this manuscript and the constructive suggestions to benchmark findings with the previous studies. In the revision, we added the comparison to these three publications, as well as the main differences between MiSplice and SAVnet (see the sentences in red on page 5). In summary, we found 150 novel nc-SCMs from WGS and 178 novel nc-SCMs from WES, which are more than half of the ~600 events described in the paper. We also updated supplementary Tables 1 and 2 to specifically indicate which non-coding splice-site-creating mutations are unique to our study and which ones were reported in previous publications. In addition, the PCAWG Transcriptome Core Group bioRxiv 2018 work was published in Nature, 578, 129–136 (2020), so we have updated this citation.

MAJOR

1. P. 5: When filtering their identified nc-SCMs, they reduced false positives by filtering splicing events observed in at least 10 control samples. How did they determine the threshold of 10 control samples?

The motivation to filter sites with less than 10 control samples is because one cancer type, DLBC, has only 7 samples with both WGS mutation calls and RNA-Seq data (see figure below). In light of the reviewer's comment, we felt that it is better to remove this cancer type (DLBC), thus obviating the need for filtering in downstream analyses. In the revision, we have removed DLBC, and reduced the sample size from 790 to 783 for TCGA WGS data included in the study. Importantly, the analysis results did not change. We have made the necessary changes in Figure 1 and added details under Data Availability.

2. P.5: They also state that “sites with spliced-in mutations were required to have at least 30% supporting reads...”. It is unclear what are “sites with spliced-in mutations”. Does that mean the mutation was observed in the RNA-Seq read? What is the basis of this specific filter?

We apologize for not making it clear in the initial submission. Yes, spliced-in mutations mean that they were observed in the RNA-Seq reads. We added text to clarify the definition and why we use this specific filter; see pages 5 and 6.

3. Exon creation is described in the PCAWG study and should be referenced.

We thank the reviewer for pointing out the omission. We have cited the PCAWG study in the revision; see the second paragraph of page 6.

4. P.6, starting at line 167, the authors characterize new exons that “often” carry a stop codon or that “most” complex events involve concurrent exon shrinkage. Exactly how many of these are there?

We again thank the reviewer for pointing out the omission. We have added these specific numbers in the revision; see the first paragraph of page 7.

5. The authors highlight a CAND2-RPL32 fusion transcript that is associated with a splice site creating mutation. Should this be written as RPL32-CAND2? Based on the Supp Figure and that the mutation creates a 3' splice site, I would assume that fusion transcript would have RPL32 at the beginning, followed by CAND2.

We apologize for the oversight in naming this fusion. In the revised version, we have changed “CAND2-RPL32” to “RPL32-CAND2”.

6. Line 199 states that 28 genes had expression outliers, but Figure 5 shows 30 and the legend also states there are 30.

We apologize for the typo and have corrected it in the revised version. As suggested by Reviewer 2, we added FDR correction for all the expression outliers, and found that they did not pass the $FDR < 0.05$ cut-off. Thus, we decided to move the outliers subpanel to supplementary Fig. 6.

7. Regarding the mini-gene assays, it is unclear why the particular 5 genes were selected for validation. Also, for some of the mutant products shown in their gels on Fig. 7A, it is very difficult to resolve a difference in products size. For example, the EP300 mutant product is 9bp different from the wild-type product. The gels shown cannot resolve this difference. They look like the same size. Moreover, I'm unclear what is exactly shown in the Sanger sequencing validation shown in Supplementary figures. The legends say it highlights the plasmid boundary. Wouldn't you want to sequence the mutant junction product? And if so, this would be difficult to do given the poor resolution on the gel.

We thank the reviewer for these insightful points. We selected these five genes based on their important roles in cancer initiation and progression (noted in the last paragraph of page 9 of the revised manuscript). While the resulting product size differences for smaller events cannot be visually resolved on the gel, we extracted DNA from the wild type and mutant products and performed Sanger sequencing to validate our predictions. Now we also show the RT-PCR sequence expected when using the pCAS2.1 plasmid for the mini-gene splicing assay in the "minigene splicing assay" section of methods. Within the predicted RT-PCR sequence we highlighted the position where the inserted new exon sequence should be present if the mutant product were expressed ("CTGGCTGGGGATAGAGCCT"), specifically within the two underlined and bolded "**GG**" nucleotides representing the boundary between the two endogenous exons in the pCAS plasmid. For the EP300 mutant for example, the exon of the endogenous exon from the pCAS2.1 plasmid is highlighted (GGCTGGG) and directly to the right we can confirm that the expected mutant product is TTTGTATAG (refer to Figure 6b - also highlighted in image below for simplicity).

We hope this description helps to clarify how we validated the novel mutant products. In the revised manuscript, we updated supplementary Fig. S7-S10 by adding the schematics of the minigene assay. In these schematics, we showed the exon of the pCAS2.1 plasmid and the novel exon created by mutations to make the results more interpretable.

8. In their discussion, they should expand a bit on their observation that they see more cases of increased outlier expression of the genes rather than a decrease in outlier expression. This perhaps mean that there are additional nc-SCMs that are missed because the splice product is degraded and cannot be detected by observing the splice junction.

We have added this point to the discussion; see the marked sentences on page 10.

9. Line 283: They claim this is the first study to demonstrate that 20% of tumors harbor mutations that disrupt splicing patterns. Again, how does this compare to their own 2018 study and the Shiraishi study?

We have added the comparison to our own 2018 study and the Shiraishi study, as mentioned above. With respect to this particular point, those respective studies reported 1% and 9%, which are appreciably smaller figures than what we report here. We have added text regarding this aspect to the last paragraph of the discussion (page 10).

10. Figure 2C: I'm not sure what the black line is supposed to indicate. The black line on the violin plot for the "New exon" is at a location I wouldn't expect. Usually these are at the median value, but it doesn't appear to be on the plot. Also, there is a letter "A" typo over the "affected genes" violin plot. Perhaps this is a PDF conversion issue.

We apologize for the confusion and typo. The black lines indeed denote the median, but the "New exon" line was misplotted. It is now fixed (See updated figure below). We have removed the redundant "A" in the revised figure.

11. Figure 7B: It is unclear what the red line indicates? It seems to be placed on the created splice site for EP300 and KMT2B, but for the other two genes, the red line seems to be at a random location.

We apologize for the confusion. These red lines indicate mutation positions, which was not clear in the original submission. We have clarified it in the figure caption.

MINOR

1. Instances of “splicing site” should be changed to “splice site”:
 - a. In the title
 - b. Also subheader on p.8 should be changed from “non-coding splicing-creating” to “splice-creating”.
 - c. Line 236: “alternative splicing sites” changed to “alternative splice sites”

We have made these recommended changes. Extra care was taken to ensure that all such instances have been corrected throughout the manuscript.

2. P.3, last sentence: “alternative splicing phenomena enriched to non-coding regions” should be changed to ““alternative splicing phenomena enriched in non-coding regions”

We have made this change.

3. P. 4, line 86: Remove the word “respectively”

We have removed it.

4. line 94: There is an author’s comment that should be removed

We apologize for leaving this errant comment in the manuscript. We have removed it.

5. line 96: “perform” should be changed to “performance”

This has been corrected.

6. line 117: “no-coding” should be changed to “non-coding”

This was a typo and has been corrected.

7. line 268 mentions this is ICGC WGS. Do they mean TCGA WGS?

We thank the reviewer for pointing this out. We have changed ICGC to TCGA.

8. I find Figure 4 to be somewhat uninformative. If there is a need to reduce the number of figures, I would suggest this be moved to supplement.

We thank the reviewer for the suggestion and have moved Figure 4 to the supplement. It is now labeled as supplementary Fig. 4.

Reviewer #2 (Remarks to the Author)

Review comments:

The authors use available WGS, WXS and RNAseq data to identify somatic mutations which potentially cause splicing alterations in cis. Their method aims at discovering a high confidence set of 'novel' junctions near (~20nt) splice site creating mutations, integrating various parameters like supporting reads per mutation, junction, mapping quality and filtering against canonical junctions. The authors find a reasonable sized set of candidate mutations that track well with observed splicing alterations and then aim to characterize general impact patterns these mutations have on splicing. Furthermore, the authors investigate transcript changes in some target genes of interest (in this case cancer related genes). Overall, this manuscript shows the utility of integrating WGS and RNAseq data to stratify somatic mutations in non-coding regions and gives some insight how even non-coding mutations can be functionally relevant in disease by altering splicing patterns.

We thank the reviewer for the appreciation of our work on discovering non-coding splice-site-creating mutations in cancer.

Major points:

- How exactly were the 281 splice altering mutations manually reviewed in IGV? What was assessed and what criteria were used? This needs at least some section in methods and clarification/justification why there was a need for manual review.

We thank the reviewer for pointing this out. In the revised manuscript, we added the "IGV manual review" section in the methods to show how and why we did the manual review; see sentences marked in red on page 14.

- Fig3A enrichment in mutations +/- 3 bp from SS, the data presented do not show that. Donors have enrichments near the SS in both intron and exon, whereas acceptors have a peak at -1 (but not as the authors claim -3).

We apologize for the confusion in the manuscript. Now we have corrected it in the revised version; see sentence marked in red on page 6.

o Mutations that destroy the canonical AG and GU motif have already been annotated (e.g. ensemble's variant effect predictor) – the authors should address how many of their non-coding variants are actually altering splicing beyond destruction of canonical splice sites

We agree this was not clear in the first submission. For the non-coding splice-site-creating mutations curated in this study, we removed mutations that destroy the canonical AG and GU motif, which were annotated as splice-site mutations in the MAF file. Specifically, we have removed "Silent", "Splice_Site", "Misense_Mutation", "Nonsense_Mutation", "Nonstop_Mutation", "In_Frame_Del", "In_Frame_Ins", "Frame_Shift_Del", and

“Frame_Shift_Ins” variants, which are reported as coding mutations in previous studies in order to focus our examination on non-coding mutations which have not been extensively studied before. We have clarified this point on page 5 of the revised manuscript.

- The authors state: “17 out of 219 genes are associated with cancer” and later that this ‘constitutes a significant enrichment in cancer genes affected by nc mutations.’ How is that enrichment analysis performed? What is the reference gene set for cancer associated genes? COSMIC? What is the enrichment being compared to? Random gene sets? This needs further explanation in the method section. Given the extensive size of cancer associated genes, any random gene list of >200 genes will likely feature 10-20 cancer associated genes. Is this enrichment or just expectation?

We appreciate that this aspect was not clear in the initial submission. We used Fisher’s exact test to test the overrepresentation of non-coding splice-site-creating mutations found in cancer related genes. We added a sub-section to the Methods section explaining our approach; see page 14. We believe this aspect should now be clear.

- What are the JAVs and gene expression changes for cancer related genes? Why no percentage spliced in (PSI)?

We thank the reviewer for the suggestion. We added the JAVs and gene expression changes for cancer related genes and also compared them to the results from non-cancer related genes. We did not find any significant differences between cancer and non-cancer-related genes (See below). The expression change is defined as the difference of the gene expression between samples with nc-SCM and the median of samples without nc-SCM in the same cancer type.

For the latter question about PSI, JAV serves a very similar purpose for estimating the extent to which the alternative splice form is used. We used JAV here since it is readily coded in our pipeline. Future versions of MiSplice will likely offer PSI as an option.

- Also, regarding expression outliers there might be some need for multiple testing correction, given only a 5% FDR for 369 mutations spread over 33 tumor types, many reported expression outliers might be coincidence.

We thank the reviewer for this suggestion. We have added FDR correction for the outlier analyses and found that these outliers did not pass the conventional FDR cutoff (5%). We have therefore moved the outlier analyses to the supplementary Fig. 6 and added the discussion on page 8; see sentences marked in red.

- MiSplice tool is already published and no new methodological changes are made (as is also admitted in the text - see lines 94-96).

This is true. We have removed the redundant comment. In this manuscript, we focused mainly on applying the MiSplice pipeline to discover non-coding splice-site creating mutations along with some benchmark tests for the pipeline, which were not performed in our previous publication (Cell Reports, 2018).

- language sometimes quite vague (e.g., line 102: reasonably sensitivity)

We thank the reviewer for pointing this out. We have modified the language to be more precise; see Page 4.

- Power calculation is per sample. Have the authors thought about combining evidence from multiple samples to look for lower-coverage but recurrent changes, possibly increasing power.

While recurrence is often exploited to increase statistical power, there are only 2 recurrent non-coding splice-site-creating mutations in our study. Since lack of recurrence seems to be a general feature of our data, we did not consider it further. We expect it would be included in future applications of MiSplice pipeline to other data sets with more recurrent splice-site-creating mutations. We have added some text mentioning this in the last paragraph of the discussion.

- The minimal read depths shown in the power calculation of Figure 1B is 10, but the actual coverage threshold used for discover is 5 (line 122). Why?

We thank the reviewer for catching this oversight. A revised version of this plot now shows the lower limit of 5X read depth; see Figure 1.

- For the outlier analysis it is not quite clear what the distributions are over. For instance in Figure 1B - STK11 is under-expressed in HNSC. Does this only occur in HNSC? Is the distribution over all HNSC samples?

We apologize for not making it clear in the first submission. Yes, it only occurs in HNSC and the distribution is across all HNSC samples. All outlier analyses are at a cancer type level. We made it clear in the caption of supplementary Fig. 6.

Minor points:

- ISB-CGC in line 301 misses a version / access time (how were the RNA-Seq alignments generated?)

We thank the review for pointing this out. The ISB-CGC (<https://isb-cgc.appspot.com>) access of TCGA RNA-seq bams was granted through tcga-phs000178-controlled credential. TCGA RNA-Seq alignments used in the current study were generated using MapSplice against the human hg19 reference. The complete commands and references to replicate the TCGA RNA-Seq bams can be found from https://webshare.bioinf.unc.edu/public/mRNAseq_TCGA/. We have added these details in the Dataset Description section.

- line 356: Is the correction applied across all sites or what is referred to by trials?

The word “trial” is commonly associated with a single hypothesis test, so “trial” here is equivalent to “site”, i.e. correction is applied across all sites. We apologize for the confusion and have changed “trial” to “site” for clarity.

- Figure 2C: Do the asterisks refer to different significance levels?

We apologize for not explicitly indicating what the asterisks refer to. They denote different p values as following:

*: $P \leq 0.05$

** : $P \leq 0.01$

***: $P \leq 0.001$

****: $P \leq 0.0001$

We now clearly indicate these values in Figure 2’s caption.

- Image quality of minigene reporter agarose gels

o Why are there varied size PCR amplicons in MUT but not WT PCR for MAPK, KMT2B and BCOR?

The additional band in the mutant product is seen at ~235 bp which is the endogenous exons in the pCAS2.1 plasmid without the mutant exon. Often with this assay, if the new mutant splice site is not strong enough, we see multiple alternatively spliced products for some of the mutant constructs. This is an expected observation with the pCAS2.1 plasmid and the mini-gene splicing assay since the endogenous exons have very strong splice sites. We had a similar observation in our 2018 coding splice-site-creating paper (Jayasinghe et al., Cell reports 2018, 23:270-281). We have added these discussions on page 9.

Typos:

* Figure 2C: axis label: Log lenth  Log length

* Figure 3A: axis label: Spice Site  Splice Site

We apologize for the typos and have corrected them in the revision.

Reviewer #1 (Remarks to the Author):

I think the author sufficiently addressed most of my previous concerns. However, my only major remaining concern is that I do not think the comparison with the previously published tool, SAVnet, has been thoroughly performed or presented. How does SAVnet compare with MiSplice in terms of both sensitivity and specificity? On p. 5, the authors state that there is a 33% overlap between SAVnet SAVs discovered from WGS and nc-SCMs identified through MiSplice and that the main difference between SAVnet and MiSplice is that "the latter does not restrict somatic variants to the specific positions of newly created splice-sites". Based on their manual review of 281 nc-SCMs, they estimate 19% are false positives. Did these false positives tend to correspond to mutations that fall outside of the typical splice site positions that SAVnet examines? For the SAVnet-specific SAVs that MiSplice did not identify, do those tend to be false-positives? Why did MiSplice not detect those cases?

The following are minor comments.

p. 5, line 118: on-coding mutations should be changed to non-coding mutations

p. 5, line 125: The wrong citation is given for the PCAWG Consortium paper

p.8, line 222: The STK11 new exon was highlighted in the PCAWG Consortium paper and should be cited.

Reviewer #2 (Remarks to the Author):

The concerns previously raised have been addressed.

Responses to Reviewers

Reviewer #1 (Remarks to the Author)

I think the author sufficiently addressed most of my previous concerns. However, my only major remaining concern is that I do not think the comparison with the previously published tool, SAVnet, has been thoroughly performed or presented. How does SAVnet compare with MiSplice in terms of both sensitivity and specificity? On p. 5, the authors state that there is a 33% overlap between SAVnet SAVs discovered from WGS and nc-SCMs identified through MiSplice and that the main difference between SAVnet and MiSplice is that “the latter does not restrict somatic variants to the specific positions of newly created splice-sites”. Based on their manual review of 281 nc-SCMs, they estimate 19% are false positives. Did these false positives tend to correspond to mutations that fall outside of the typical splice site positions that SAVnet examines? For the SAVnet-specific SAVs that MiSplice did not identify, do those tend to be false-positives? Why did MiSplice not detect those cases?

We are pleased that the reviewer feels we have sufficiently addressed most of the previous comments. Regarding the one remaining concern, we have added a more detailed comparison with SAVnet for WGS nc-SCMs in the revised manuscript, together with more detail on the methodological differences between the two pipelines; See marked sentences in red on pages 5 and 6. Below is a list of responses to specific comments, which can also be found on the marked sentences of pages 5 and 6.

1. How does SAVnet compare with MiSplice in terms of both sensitivity and specificity?

Based on the TCGA WGS data, we estimated the respective sensitivity and specificity for SAVnet as roughly 46% and 42% and MiSplice as 87% and 81%.

2. On p. 5, the authors state that there is a 33% overlap between SAVnet SAVs discovered from WGS and nc-SCMs identified through MiSplice and that the main difference between SAVnet and MiSplice is that “the latter does not restrict somatic variants to the specific positions of newly created splice-sites”. Based on their manual review of 281 nc-SCMs, they estimate 19% are false positives. Did these false positives tend to correspond to mutations that fall outside of the typical splice site positions that SAVnet examines.

This is a good point. For nc-SCMs identified by MiSplice, we indeed saw a high false-positive rate (25%) for mutations that SAVnet does not report. For nc-SCMs reported by both SAVnet and MiSplice, we observed a low false-positive rate (~5%).

3. For the SAVnet-specific SAVs that MiSplice did not identify, do those tend to be false-positives? Why did MiSplice not detect those cases?

For nc-SCMs uniquely predicted by SAVnet, we indeed found a high false-positive rate of about 78% by IGV manual review. Through the comparison of SAVnet and MiSplice results, we

noticed that some true newly created splice sites could be far away from the mutation when the mutation is a nearby position of the canonical splice site, which functions as disrupting the canonical ss. SAVnet captures this scenario as long as the mutation is close to the canonical ss. However, since MiSplice implements a 20bp cut-off between mutation and the newly created ss, it misses these unique calls predicted by SAVnet. We added a more detailed description of the results on page 5 and 6.

The following are minor comments.

p. 5, line 118: on-coding mutations should be changed to non-coding mutations

We thank the reviewer for pointing out the typos, and we have fixed the typo on page 5.

p. 5, line 125: The wrong citation is given for the PCAWG Consortium paper

We have likewise fixed this.

p.8, line 222: The STK11 new exon was highlighted in the PCAWG Consortium paper and should be cited.

We thank the reviewer for pointing out the omission. We have added the citation for the PCAWG Consortium paper for the STK11 new exon on page 8.

Reviewer #2 (Remarks to the Author):

The concerns previously raised have been addressed.

We thank the reviewer for reviewing our manuscript and appreciating that we have addressed the previous concerns.

Reviewer #1 (Remarks to the Author):

The authors have addressed all my remaining concerns and the differences between their method and previous ones have been clarified.